# Spin-phonon relaxation from a universal ab initio density-matrix approach

Junqing Xu[1,3], Adela Habib[2,3], Sushant Kumar[2], Feng Wu[1], Ravishankar Sundararaman[2✉] & Yuan Ping [1✉]

Designing new quantum materials with long-lived electron spin states urgently requires a general theoretical formalism and computational technique to reliably predict intrinsic spin relaxation times. We present a new, accurate and universal first-principles methodology based on Lindbladian dynamics of density matrices to calculate spin-phonon relaxation time of solids with arbitrary spin mixing and crystal symmetry. This method describes contributions of Elliott-Yafet and D'yakonov-Perel' mechanisms to spin relaxation for systems with and without inversion symmetry on an equal footing. We show that intrinsic spin and momentum relaxation times both decrease with increasing temperature; however, for the D'yakonov-Perel' mechanism, spin relaxation time varies inversely with extrinsic scattering time. We predict large anisotropy of spin lifetime in transition metal dichalcogenides. The excellent agreement with experiments for a broad range of materials underscores the predictive capability of our method for properties critical to quantum information science.

[1] Department of Chemistry and Biochemistry, University of California, Santa Cruz, CA 95064, USA. [2] Department of Materials Science and Engineering, Rensselaer Polytechnic Institute, 110 8th Street, Troy, New York 12180, USA. [3]These authors contributed equally: Junqing Xu, Adela Habib. ✉email: sundar@rpi.edu; yuanping@ucsc.edu

The manipulation of electron spins is of increasing interest in a wide-range of emerging technologies. The rapidly growing field of spintronics seeks to control spin as the unit of information instead of charge in devices such as spin transistors[1]. Quantum information technologies seek to utilize localized spin states in materials both as single-photon emitters[2–4] and as spin-qubits for future integrated quantum computers[5]. Both spintronics and quantum information applications therefore demand a quantitative understanding of spin dynamics and transport in metals and semiconductors. Recent advances in circularly polarized pump-probe spectroscopy[6], spin injection, and detection techniques[7] have enabled increasingly detailed experimental measurement of spin dynamics in solid-state systems[8,9]. However, a universal first-principles theoretical approach to predict spin dynamics, quantitatively interpret these experiments and design new materials has remained out of reach.

A key metric of useful spin dynamics is the spin relaxation time $\tau_s$[1]. For example, spin-based quantum information applications require $\tau_s$ exceeding milliseconds for reliable qubit operation. Consequently, accurate prediction of $\tau_s$ in general materials is an important milestone for first-principles design of quantum materials. Spin–spin[10], spin–phonon[11], and spin-impurity scatterings, all contribute to spin relaxation, but spin–phonon scattering sets the intrinsic material limitation and is typically the dominant mechanism at room temperature[1]. Further, spin–phonon relaxation arises from a combination of spin–orbit coupling (SOC) and electron–phonon scattering, and is traditionally described by two mechanisms. First, the Elliott–Yafet (EY) mechanism involves spin–flip transitions between pairs of Kramers-degenerate states due to SOC-based spin-mixing of these states[12,13]. Second, the D'yakonov–Perel' (DP) mechanism in systems with broken inversion symmetry involves electron spins precessing between scattering events due to the SOC-induced internal effective magnetic field[14].

Previous theoretical approaches have extensively investigated these two distinct mechanisms of intrinsic spin–phonon relaxation using model Hamiltonians in various materials[15]. These methods require parametrization for each specific material, which needs extensive prior information about the material and specialized computational techniques, and often only studies one mechanism at a time. Furthermore, most of these approaches require the use of simplified formulae[12,14] and make approximations to the electronic structure (e.g. low spin-mixing) or electron–phonon matrix elements[15]. This limits the generality and reliability of these approaches for complex materials, particularly for the DP mechanism, where various empirical relations are widely employed to estimate $\tau_s$[1]. Sophisticated methods based on spin susceptibility[16] and time evolution of density matrix[17] also rely on suitably chosen model Hamiltonians with empirical scattering matrix elements. Therefore, while these methods provide some mechanistic insight, they do not serve as predictive tools of spin relaxation time for the design of new materials.

A general first-principles technique to predict spin–phonon relaxation in arbitrary materials is therefore urgently needed. Previous first-principles studies have addressed the EY mechanism in centrosymmetric semiconductors[18,19] and metals[20]. These methods[18,20] rely on defining a pseudospin that allows the use of Fermi's golden rule (FGR) with only spin–flip transitions[13]. However, this is only well-defined for cases with weak spin-mixing such that eigenstates within each Kramers-degenerate pair can be chosen to have small spin-minority components, precluding the study of spin relaxation of states with strong spin-mixing, e.g. holes in silicon and noble metals. First-principles calculations have not yet addressed systems with such complex degeneracy structures, where the simple picture of spin–flip matrix elements in a FGR breaks down, or systems without

inversion symmetry that do not exhibit Kramers degeneracy. Therefore, a more general first-principles technique without the material-specific simplifying assumptions of these previous approaches is now necessary.

In this work, we establish a new, accurate and unified first-principles technique for predicting spin relaxation time based on perturbative treatment of the Lindbladian dynamics of density matrices[21]. Importantly, by covering previously disparate mechanisms (e.g. EY and DP) in a unified framework, this technique is applicable to all materials regardless of dimensionality, symmetry (especially inversion) and strength of spin-mixing, which is critical for new material design. All SOC effects are included self-consistently (and non-perturbatively) in the ground-state eigensystem at the density functional theory (DFT) level, and we predict $\tau_s$ through a universal rate expression without the need to invoke real-time dynamics. In this article, we first introduce our theoretical framework based on first-principles density-matrix dynamics, and then show prototypical examples of $\tau_s$ for the broad range of systems, including three with inversion symmetry—silicon, iron, and graphene, and three without inversion symmetry—monolayer $MoS_2$, monolayer $MoSe_2$, and bulk GaN, in excellent agreement with available experimental data. By doing so, we establish the foundation for quantum dynamics of open systems from first-principles to facilitate the design of quantum materials.

## Results

**Theory**. The key to treating arbitrary state degeneracy and spin-mixing for spin relaxation is to switch to an ab initio density-matrix formalism, which goes beyond specific cases such as Kramers degeneracy or Rashba-split model Hamiltonians. Specifically, we seek to work with density matrices of electrons alone, treating its interactions with an environment consisting of a thermal bath of phonons. In general, tracing out the environmental degrees of freedom in a full quantum Liouville equation of the density-matrix results in a quantum Lindblad equation. Specifically, for electron–phonon coupling[21] based on the standard Born–Markov approximation[22] that neglects memory effects in the environment, the Lindbladian dynamics in interaction picture reduces to

$$\frac{\partial \rho_{\alpha_1 \alpha_2}}{\partial t} = \frac{2\pi}{\hbar N_q} \mathrm{Re} \sum_{q\lambda\pm\alpha'\alpha_1'\alpha_2'} \left[ \begin{array}{c} (I-\rho)_{\alpha_1\alpha'} \left(G^{q\lambda\pm}\right)_{\alpha'\alpha_1'} \rho_{\alpha_1'\alpha_2'} \left(G^{q\lambda\mp}\right)_{\alpha_2'\alpha_2} \\ - \left(G^{q\lambda\mp}\right)_{\alpha_1\alpha'}(I-\rho)_{\alpha'\alpha_1'} \left(G^{q\lambda\pm}\right)_{\alpha_1'\alpha_2'} \rho_{\alpha_2'\alpha_2} \end{array} \right] n_{q\lambda}^{\pm},$$

(1)

where $\alpha$ is a combined index labeling electron wavevector $k$ and band index $n$, $\lambda$ is mode index and $\pm$ corresponds to $q = \mp(k - k')$. $n_{q\lambda}^{\pm} \equiv n_{q\lambda} + 0.5 \pm 0.5$ and $n_{q\lambda}$ is phonon occupation. $G_{\alpha\alpha'}^{q\lambda\pm} = g_{\alpha\alpha'}^{q\lambda\pm} \delta^{1/2}(\varepsilon_\alpha - \varepsilon_\alpha' \pm \omega_{q\lambda})$ is the electron–phonon matrix element including energy conservation, where $\omega_{q\lambda}$ is the phonon frequency.

This specific form of the Lindbladian dynamics preserves positive definiteness of the density matrix which is critical for numerical stability[21]. In addition, the energy-conserving $\delta$-function above is regularized by a Gaussian with a width $\gamma$, which corresponds physically to the collision time. In some cases, the results depend on $\gamma$ and $\gamma \rightarrow 0$ is not the relevant limit[23]. Here, the Lindblad master equation with finite smearing parameters corresponding to the collision time can be regarded as the best Markovian approximation to the exact dynamics[23]. In the case of spin relaxation, this is particularly important for systems that exhibit the DP mechanism, as we show below. Consequently, we consistently determine the smearing parameters from ab initio electron–phonon linewidth calculations throughout[24,25].

The density-matrix formalism allows the computation of any observable such as number and spin density of carriers, and the inclusion of different relaxation mechanisms at time scales spanning femtoseconds to microseconds, which forms the foundation of the general relaxation time approach we discuss below. Given an exponentially relaxing measured quantity $O = \mathrm{Tr}(o\rho)$, where $o$ and $\rho$ are the observable operator and the density matrix, respectively, we can define the relaxation rate $\Gamma_o$ and relaxation time $\tau_o = \Gamma_o^{-1}$ of quantity $O$ as

$$\frac{\partial(O - O^{\mathrm{eq}})}{\partial t} = -\Gamma_o(O - O^{\mathrm{eq}}), \tag{2}$$

where eq corresponds to the final equilibrium state. We note that even when the observables have additional $\cos(\omega t)$ oscillation factors, such as due to spin precession with periodicity of $\omega$, the above equation remains an appropriate definition of the overall relaxation rate. For example, for a precessing and relaxing spin system with $S(t) = S_0 \exp(-t/\tau)\cos(\omega t)$, the initial relaxation rate is $\dot{S}(0) = -S_0/\tau$, which is exactly the same as that of a pure exponential relaxation.

The equilibrium density matrix in band space is $(\rho^{\mathrm{eq}})^k_{nn'} = f_{kn}\delta_{nn'}$, where $f_{kn}$ are the Fermi occupation factors of electrons in equilibrium. Writing the initial density matrix $\rho = \rho^{\mathrm{eq}} + \delta\rho$, assuming a small perturbation $\|\delta\rho\| \ll \|\rho^{\mathrm{eq}}\|$ and $k$-diagonal $o$ and $\delta\rho$, the Lindblad dynamics expression (Eq. (1)) and the definition (Eq. (2)) yield

$$\Gamma_o = \frac{2\pi}{\hbar N_q \mathrm{Tr}(o\delta\rho)} \mathrm{Tr}_n \mathrm{Re} \sum_{kk'\lambda} \left[ o, G^{q\lambda-} \right]_{kk'}$$
$$\times \begin{bmatrix} (\delta\rho)_k G_{kk'}^{q\lambda-}(n_{q\lambda} + I - f_{k'}) \\ - (n_{q\lambda} + f_k) G_{kk'}^{q\lambda-}(\delta\rho)_{k'} \end{bmatrix}^{\dagger_n}. \tag{3}$$

Here, the $G$ is exactly as defined above in Eq. (1), but separating the wavevector indices $(k, k')$ and writing it as a matrix in the space of band indices $(n, n')$ alone. Similarly, $o$ and $\delta\rho$ are also matrices in the band space, $\mathrm{Tr}_n$ and $\dagger_n$ are trace and Hermitian conjugate in band space, and $[o, G]_{kk'} \equiv o_k G_{kk'} - G_{kk'} o_{k'}$, written using matrices in band space.

Given an initial perturbation $\delta\rho$ and an observable $o$, Eq. (3) can now compute the relaxation of expectation value $O$ from its initial value. Even for a specific observable like spin, several choices are possible for the initial perturbation corresponding directly to the experimental measurement scheme. Specifically for spin relaxation rate $\Gamma_{s,i}$, the observable is the spin matrix $S_i$ labeled by Cartesian directions $i = x, y, z$, and the initial perturbed state should contain a deviation of spin expectation value from equilibrium. The most general (experiment-agnostic) choice for preparing a spin polarization is to assume that all other degrees of freedom are in thermal equilibrium, which can be implemented using a test magnetic field $B_i$ as a Lagrange multiplier for implementing a spin polarization constraint. With a corresponding initial perturbation Hamiltonian of $H_1 = -2\mu_{\mathrm{B}}B_i S_i/\hbar$, where $\mu_{\mathrm{B}}$ is the Bohr magneton, perturbation theory yields

$$\delta\rho_{k,mn} = -\frac{2\mu_{\mathrm{B}}B_i}{\hbar}\frac{f_{km} - f_{kn}}{\epsilon_{km} - \epsilon_{kn}}S_{i,k,mn}. \tag{4}$$

In some cases, $S_{i,k,mn} \approx 0$ when $\epsilon_{km} \neq \epsilon_{kn}$. For these cases, $\delta\rho \approx -(2\mu_{\mathrm{B}}B_i/\hbar)(\partial f/\partial\epsilon)S_i^{\mathrm{deg}}$, where $(S_i^{\mathrm{deg}})_{knn'} \equiv (S_i)_{knn'}\delta_{\epsilon_{kn}\epsilon_{kn'}}$ is the degenerate-subspace projection of $S_i$. In such cases, we can

further simplify Eq. (3) to the Fermi Golden rule-like expression,

$$\Gamma_{s,i} = \frac{2\pi}{\hbar N_k N_q k_{\mathrm{B}} T \chi_{s,i}} \sum_{kk'\lambda \pm nn'} \left\{ \left| \left[ S_i^{\mathrm{deg}}, g^{q\lambda-} \right]_{knk'n'} \right|^2 \right.$$
$$\left. \delta(\epsilon_{kn} - \epsilon_{k'n'} - \omega_{q\lambda})f_{k'n'}(1 - f_{kn})n_{q\lambda} \right\}, \tag{5}$$

where $\chi_{s,i} = \mathrm{Tr}_n[S_i(-\partial f/\partial\epsilon)S_i^{\mathrm{deg}}]/N_k$. Note that the test field $B_i$ etc. drops out of the final expression and only serves to select the direction of the perturbation in the high-dimensional space of density matrices.

Without SOC, $S_i^{\mathrm{deg}} = S_i$ commutes with $g$, leading to $\Gamma_{s,i} = 0$ as expected. If $S_i^{\mathrm{deg}}$ is diagonal, $[S_i^{\mathrm{deg}}, g^{q\lambda-}]_{knk'n'}$ reduces to $\Delta s_{i,knk'n'}g_{knk'n'}^{q\lambda-}$, where $\Delta s_{i,knk'n'} \equiv s_{i,kn} - s_{i,k'n'}$ is the change in (diagonal) spin expectation value for a pair of states. Therefore, in this limit, Eq. (5) reduces to transitions between pairs of states, each contributing proportionally to the square of the corresponding spin change.

See Supplementary Note 1 and 2 for detailed derivations of the above equations. As we show in Supplementary Note 1 and 2, the above equations can be reduced to previous formulae with spin–flip matrix elements in Kramers-degenerate subspaces for systems with inversion symmetry and weak spin-mixing, such as conduction electron spin relaxation in bulk Si, similar to ref. [18]. However, Eq. (3) is much more general, applicable for systems with arbitrary degeneracy and crystal symmetry, and we therefore use it throughout for all results presented below. In addition, the overall framework can also be extended to other observables and can be made to correspond to specific measurement techniques that prepare a different initial density matrix e.g. a circularly polarized pump pulse.

Finally, note that in our first-principles method, all SOC-induced effects (such as the Rashba/Dresselhaus effects) are self-consistently included in the ground-state eigensystem or the unperturbed Hamiltonian $H_0$. This is essential to allow us to simulate $\tau_s$ by a single rate calculation when there is broken inversion symmetry. On the other hand, if SOC does not enter into $H_0$, as in previous work with model Hamiltonians, it must be treated as a separate term that provides an internal effective magnetic field. Consequently, those approaches require a coherent part of the time evolution to describe the fast spin precession induced by this effective magnetic field, which require explicit real-time dynamics simulations even to capture spin relaxation, going beyond a simple exponential decay as in Eq. (2). Using fully self-consistent SOC in a first-principles method is therefore critical to avoid this system-specific complexity and arrive at the universal approach outlined above.

**Systems with inversion symmetry: Si and Fe.** We first present results for systems with inversion symmetry traditionally described by a Elliot–Yafet spin–flip mechanism. Figure 1a shows that our predictions of electron spin relaxation time ($\tau_s$) of Si as a function of temperature are in excellent agreement with experimental measurements[26,27]. Note that previous first-principles calculations[18] approximated spin–flip electron–phonon matrix elements from pseudospin wavefunction overlap and spin-conserving electron–phonon matrix element, effectively assuming that the scattering potential varies slowly on the scale of a unit cell; we make no such approximation in our direct first-principles approach. Importantly, this allows us to go beyond the doubly degenerate Kramers-degenerate case of conduction electrons in Si. In contrast, holes in Si exhibit strong spin-mixing with spin-2/3 character and spin expectation values no longer close to $\hbar/2$. Figure 1b shows our predictions for the hole–spin relaxation time, which is much shorter than the electron case as a result of the

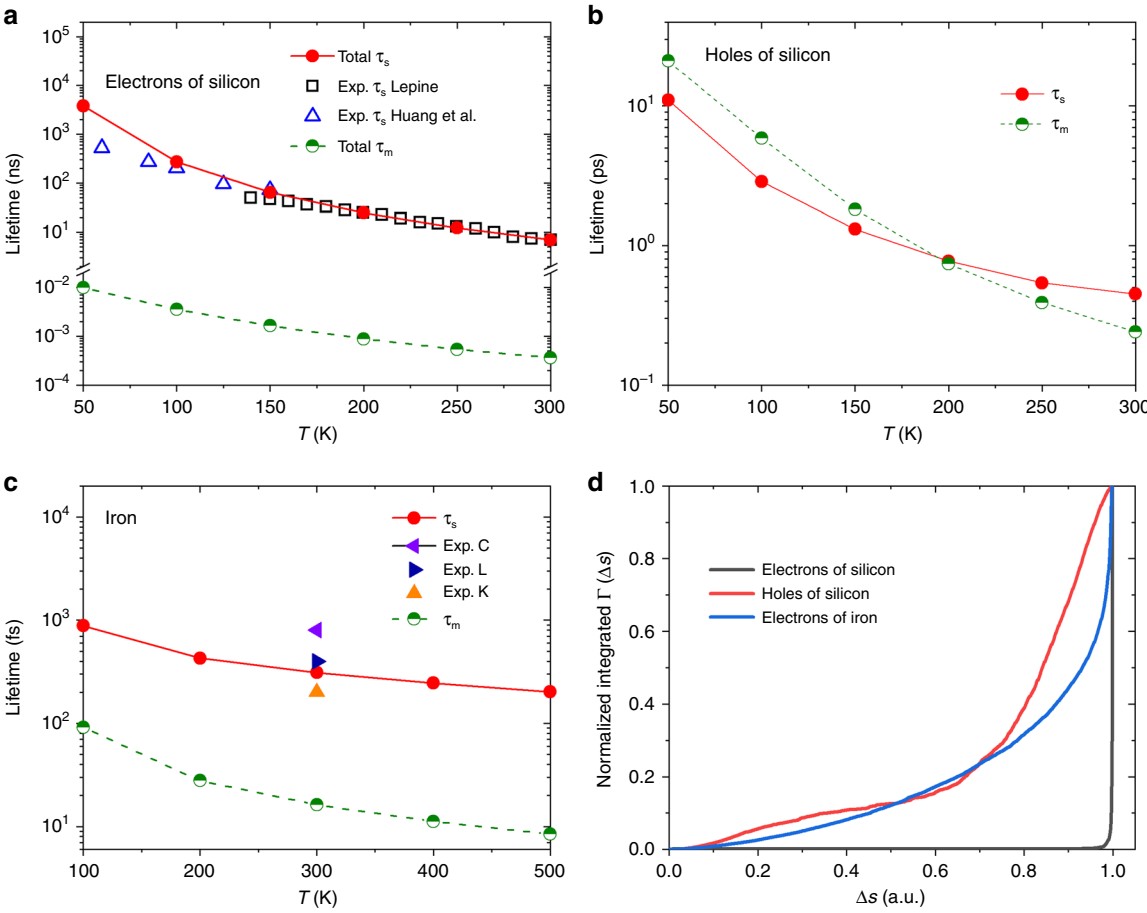

**Fig. 1 Relaxation time predictions for two systems with inversion symmetry.** Spin ($\tau_s$) and momentum ($\tau_m$) relaxation times are shown for: **a** electrons in $n$-Si with carrier concentration $7.8 \times 10^{15}$ cm$^{-3}$ (compared to experiment[26,27]), **b** holes in p-Si with carrier concentration $1.3 \times 10^{15}$ cm$^{-3}$, and **c** iron (compared to experiment[29–31]). **d** Cumulative contributions to spin relaxation by change in spin, $\Delta s$, per scattering event defined based on Eq. (5): electrons in Si exhibit spin flips with all contributions at $\Delta s = 1$, whereas holes in Si and electrons in iron exhibit a broad distribution in $\Delta s$.

strong mixing (450 fs for holes compared to 7 ns for electrons at 300 K) and is much closer to the momentum relaxation time. In addition, Fig. 1d shows that the change in spin expectation values ($\Delta s$) per scattering event (evaluated using Eq. (5)) has a broad distribution for holes in Si, indicating that they cannot be described purely by spin–flip transitions, while conduction electrons in Si predominantly exhibit spin–flip transitions with $\Delta s = 1$.

We next consider an example of a ferromagnetic metal, iron, which exhibits a complex band structure not amenable for model Hamiltonian approaches. Previous first-principles calculations for ferromagnets employ empirical Elliott relation[28] or FGR formulae with spin–flip matrix elements specifically developed for metals or ferromagnets[20]. Here, we apply exactly the same technique used for the silicon calculations above and predict spin relaxation times in iron in good agreement with experimental measurements (Fig. 1c)[29–31]. Our Wannier interpolation also enables systematic and efficient Brillouin zone convergence of these predictions which were not possible previously. Similar to holes in Si, the $\Delta s$ of Fe also exhibits a broad distribution extending from 0 to $\hbar$ in the contribution to the total spin relaxation rate (Fig. 1d). Therefore, spin relaxation in transition metals are not purely spin–flip transitions, and we expect this effect to be even more pronounced in 4$d$ and 5$d$ metals with stronger SOC than the 3$d$ magnetic metal considered here. Finally, Fig. 1a–c shows that $\tau_s$ is approximately proportional to momentum relaxation

time $\tau_m$ for both Si and bcc Fe, which is expected for spin relaxation in systems with EY mechanisms[1].

**Systems with inversion symmetry: graphene.** Graphene is of significant interest for spin-based technologies, and significant recent work with model Hamiltonians seeks to identify the fundamental limits of spin coherence in graphene[32]. Estimates vary widely from theoretical estimates on the order of microseconds to experiments ranging from picoseconds to nanoseconds[33–36], with the discrepancies hypothesized to arise from faster extrinsic relaxation in experiments. However, previous model Hamiltonian studies required parametrization of approximate matrix elements, and focus on specific phonon modes (e.g. flexural modes) for spin–phonon relaxation. Here we predict intrinsic electron–phonon spin relaxation time for free-standing graphene to firmly establish the intrinsic spin–phonon relaxation limit free of specific model choices or parameters.

Figure 2 shows the predicted spin–phonon relaxation times as a function of temperature and Fermi level position. At room temperature, our calculated lifetimes are of the same magnitude (in microseconds) as previous predictions[33] indicating that faster relaxation is likely extrinsic in experiments. However, in addition to the flexural phonon mode[33,35], in-plane acoustic (A) phonon modes have a strong and non-negligible contribution, while optical modes (O) have an overall smaller effect (Fig. 2b). We also find that the ratio between in-plane and out-of-plane spin relaxation times

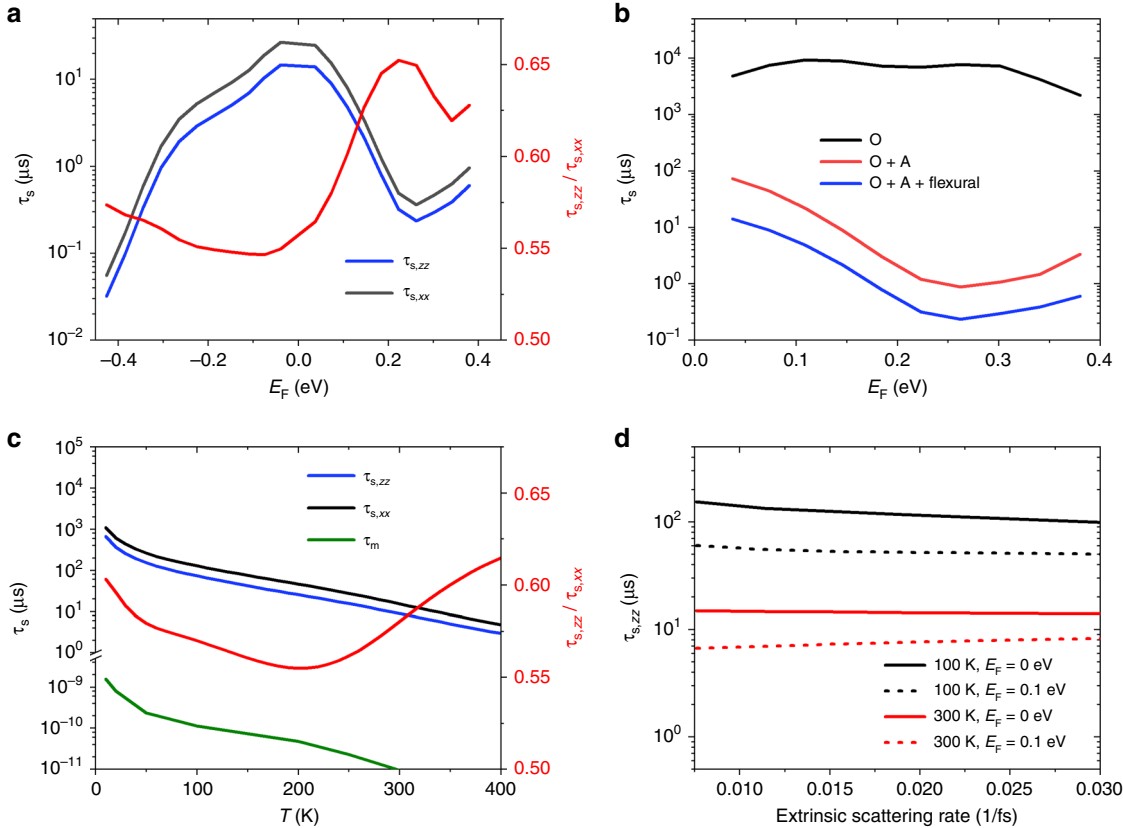

**Fig. 2 Intrinsic electron–phonon spin relaxation time of free-standing graphene.** Results are shown (**a**, **b**) at room temperature as a function of Fermi energies, **c** at $E_F = 0.1$ eV as a function of temperature and **d** as a function of extrinsic scattering rates. **b** Out-of-plane spin relaxation by cumulative phonon mode contributions starting with optical (O), adding in-plane acoustic (A), and then including the flexural mode as well, indicating the dominance of acoustic and flexural modes for spin relaxation. Both **a** and **c** show an anisotropy between in-plane and out-of-plane lifetimes with a ratio varying in the 0.5–0.7 range, whereas **d** shows that the lifetime is relatively insensitive to extrinsic scattering rates indicating that this is not the DP mechanism despite the nearly 1/2 ratio, as expected.

range from 0.5 to 0.7 (Fig. 2a, c), consistent with experimental measurements[34]. As evident from Fig. 2c, longer spin relaxation time of up to microseconds is achievable at low temperatures in pristine and free-standing graphene. However, at low temperatures, competing effects from substrates and disorder can make overall measured spin relaxation faster than theoretical predictions[35].

Finally note that while the ratio between in-plane and out-of-plane spin relaxation times is nearly 1/2, which is often considered to be a signature of the DP mechanism, free-standing graphene is inversion symmetric and does not exhibit the DP mechanism. Figure 2d shows that the spin relaxation time is mostly insensitive to the extrinsic scattering rates, instead of the linear relation (inverse relation with scattering time) expected for the DP mechanism, as discussed below in further detail. The spin relaxation of graphene may be switched to the DP regime by adding substrates or external electric fields to break inversion symmetry[36,37], which will be investigated in detail using this theoretical framework in future work.

**Systems without inversion symmetry: out-of-plane $\tau_s$ of MoS$_2$ and MoSe$_2$.** The two-dimensional transition metal dichalcogenides (TMDs) exhibit extremely long-lived spin/valley polarization (over nanoseconds)[38], with long valley-state persistence attributed to spin-valley locking effects. A fundamental understanding of spin/valley relaxation mechanisms is now required to utilize this degree of freedom for valleytronic computing[39]. Next we investigate spin relaxation $\tau_s$ of systems without inversion symmetry from first-principles, starting with two TMD systems—monolayer

MoS$_2$ and MoSe$_2$ as prototypical examples. (Unless specified, $\tau_s$ represents out-of-plane spin relaxation time $\tau_{s,zz}$ for TMDs.)

In both systems, valence and conduction band edges at $K$ and $K'$ valleys exhibit relatively large SOC band splitting, with nearly perfect out-of-plane spin polarization. Time-reversal symmetry further enforces opposite spin directions for the band-edge states at $K$ and $K'$. Previous studies using model Hamiltonians consider the DP mechanism to dominate spin relaxation in these materials[17], but in our first-principles approach, we do not need to a priori restrict our calculations to EY or DP limits.

In Fig. 3, we show the out-of-plane spin ($\tau_s$) and momentum ($\tau_m$) relaxation time of conduction electrons in two monolayer TMDs as a function of temperature, along with their intervalley/intravalley contributions and experimental values. First, the overall agreement between our calculations and previous experiments by ultrafast pump-probe spectroscopy is excellent[38,40,41]. Note that ultrafast measurements of TMDs obtain coupled dynamics of spin and valley polarizations according to the selection rules with circularly polarized light, necessitating additional analysis to extract $\tau_s$, e.g., a phenomenological model fit to experimental curves in ref. [38]. On the other hand, our first-principles method simulates $\tau_s$ directly without model or input parameters. This provides additional confidence in the experimental procedures of extracting $\tau_s$, and lends further insights into different scattering contributions in the dynamical processes as we show below. Moreover, special care is necessary when comparing with certain low temperature measurements with lightly doped samples, which access spin relaxation of excitons

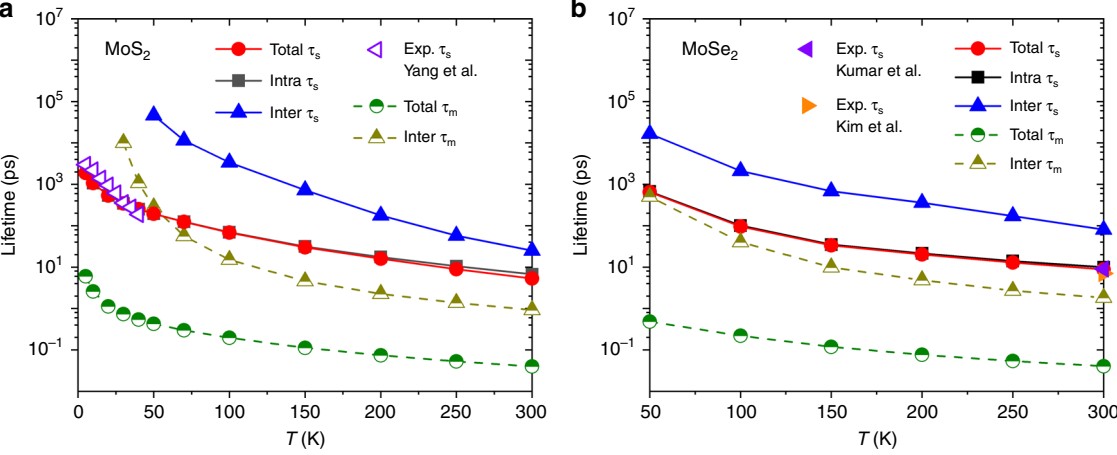

**Fig. 3 Relaxation times of conduction electrons in two systems without inversion symmetry.** Predicted spin ($\tau_s$) and momentum ($\tau_m$) relaxation times for **a** MoS$_2$ and **b** MoSe$_2$ with carrier concentrations of $5.2 \times 10^{12}$ cm$^{-2}$ and $5.0 \times 10^{11}$ cm$^{-2}$, respectively (compared to experiments[38–41]). "Intra" and "inter" denote intravalley (within $K$ or $K'$) and intervalley (between $K$ and $K'$) scattering contributions to the relaxation times; intravalley processes dominate spin relaxation at and below room temperature.

**Table 1 Percentage contributions of selected phonon modes to out-of-plane and in-plane spin relaxation time ($\tau_s$) of conduction electrons of MoS$_2$ and MoSe$_2$.**

| System | Direction | T (K) | Modes and their contributions |
|---|---|---|---|
| MoS$_2$ | Out-of-plane | 300 | ZA (12%), 1st E″ (21%), 2nd E″ (67%) |
| MoS$_2$ | In-plane | 300 | LA (55%), LO (13%) |
| MoSe$_2$ | Out-of-plane | 300 | ZA (11%), 1st E″ (36%), 2nd E″ (45%) |
| MoS$_2$ | Out-of-plane | 150 | ZA (46%), 1st E″ (24%), 2nd E″ (29%) |
| MoS$_2$ | Out-of-plane | 50 | ZA (99%) |

ZA, E″, LA and LO represent out-of-plane acoustic (flexural) mode, two lower-frequency in-plane optical modes, longitudinal acoustic and longitudinal optical phonon modes, respectively. (See phonon band structures in Supplementary Fig. 4).

rather than individual free carriers, as discussed in refs. [42,43]; we focus here on spin relaxation of free carriers.

Next, comparing the relative contributions of intervalley and intravalley scattering for spin relaxation time, we find that the intravalley process dominates spin relaxation of conduction electrons in both TMDs: the intravalley only spin relaxation time (black squares) in Fig. 3 is nearly identical with the net spin relaxation time (red circles), while the intervalley contribution alone (blue triangles) is consistently more than an order of magnitude higher in relaxation time (lower in rate). Furthermore, with decreasing temperature, the relative contribution of the intervalley process decreases because the minimum phonon energies for wave vectors connecting the two valleys exceed 20 meV, and the corresponding phonon occupations become negligible at temperatures far below 300 K.

Previous theoretical studies of MoS$_2$ with model Hamiltonians[17] obtained (out-of-plane) $\tau_s$ two orders of magnitude higher than our predictions, which agree with experimental data[38]. Such significant deviations are possibly because of the approximate treatments of electronic structure and electron–phonon coupling in their theoretical model. In addition, our first-principle calculations treat all phonon modes on an equal footing. Table 1 shows that the relative contributions of each phonon mode to $\tau_s$ varies strongly with temperature. Full electron and phonon band structure is therefore vital to correctly describe spin–phonon relaxation with varying temperature, while model Hamiltonians that select specific phonon modes have limited range of validity[17].

Hole–spin relaxation in MoS$_2$ and MoSe$_2$ has not been previously investigated in detail theoretically. Figure 4 presents our predictions of hole $\tau_s$ and $\tau_m$ in the two TMDs, indicating that hole $\tau_s$ is much longer than that for electrons at all temperatures, exceeding 1 ns below 100 K. In contrast to the electron case, the intervalley process is relatively much more important and dominates spin relaxation at low temperature in MoS$_2$ and at all temperatures in MoSe$_2$. This is because large SOC splitting at the valence band maximum makes the intravalley transition between two valence bands nearly impossible based on energy conservation in the electron–phonon scattering process. Experimental measurements also observe long spin relaxation times dominated by intervalley scattering in tungsten dichalcogenides[44], which may facilitate applications in spintronic and valleytronic devices.

External magnetic fields can serve as tools tuning material properties[45] and are an inherent component of spin dynamics measurements[38,44]. Systems with broken inversion symmetry in particular may strongly respond to magnetic fields. We therefore investigate the effects of an external field **B** on $\tau_s$ by introducing a Zeeman term $(g_s \mu_B / \hbar) \mathbf{B} \cdot \mathbf{S}$ to the electronic Hamiltonian interpolated using Wannier functions (approximating $g_s \approx 2$), just prior to computing $\tau_s$ with Eq. (3). Figure 5 shows that the out-of-plane $\tau_s$ of conduction electrons of MoS$_2$ decreases with increasing in-plane magnetic field $B_x$, in agreement with experimental work on MoS$_2$[38] and in general consistency with previous theoretical studies of $\tau_s$ for systems with broken inversion symmetry[17,46].

This strong magnetic field response has a simple intuitive explanation: in TMDs, the spin splitting of bands can be considered as the result of the internal effective magnetic field

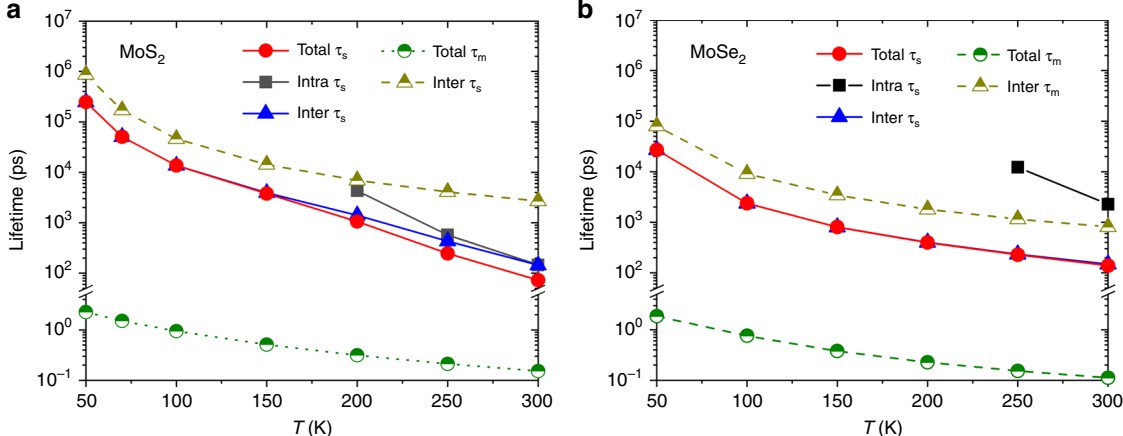

**Fig. 4 Relaxation times of holes in two systems without inversion symmetry.** Predicted spin ($\tau_s$) and momentum ($\tau_m$) relaxation times for **a** MoS$_2$ and **b** MoSe$_2$ in the low-carrier concentration limit (<$10^{11}$ cm$^{-2}$). In contrast to the electron case, the intervalley process dominates spin relaxation at low temperature in MoS$_2$ and at all temperatures in MoSe$_2$.

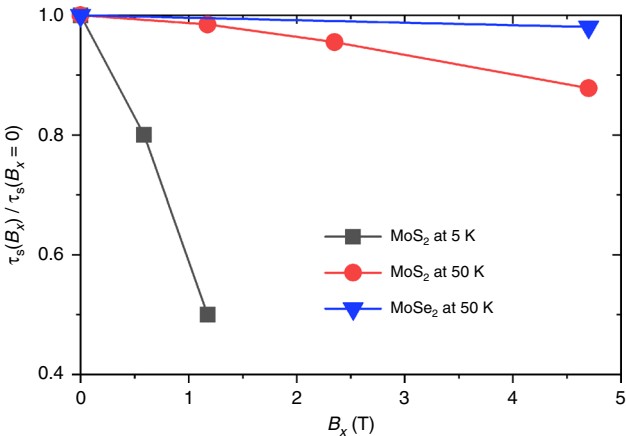

**Fig. 5 Dependence of spin relaxation on magnetic field.** Variation of spin relaxation time ($\tau_s$) with in-plane magnetic field and temperature for conduction electrons in MoS$_2$ and MoSe$_2$.

$B_{so}\hat{z}$ due to broken inversion symmetry. Applying a finite $B_x$ perpendicular to $B_{so}\hat{z}$ will cause additional spin-mixing and increase the spin–flip transition probability, thereby reducing the spin relaxation time. The degree of reduction depends on the detailed electronic structure of MoS$_2$ and MoSe$_2$ as shown in Supplementary Figs. 2 and 3: MoSe$_2$ exhibits a larger spin splitting of conduction bands and a higher internal magnetic field, and is therefore less affected by external $B_x$. Similarly, hole–spin relaxation in both MoS$_2$ and MoSe$_2$ (not shown) exhibit very weak dependence on $B_x$ because of the large spin splitting and high internal effective magnetic field $B_{so}$ for valence band-edge states compared to those near the conduction band minimum. This insensitivity of hole $\tau_s$ to magnetic fields is also consistent with experimental studies of hole $\tau_s$ in WS$_2$[44] and WSe$_2$[47].

Finally, out-of-plane magnetic field $B_z$ has a negligible effect on spin relaxation for TMDs (not shown), unlike the in-plane magnetic field $B_x$ or $B_y$. This is because electronic states around band edges are already polarized along the out-of-plane direction under a strong internal $B_{so}\hat{z}$. High experimental external magnetic fields ~1 Tesla are relatively weak in contrast and only slightly change the spin polarization of the states, rather than introducing a spin-mixing that leads to spin relaxation.

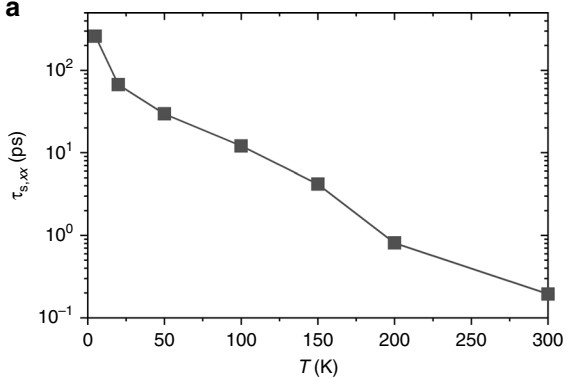

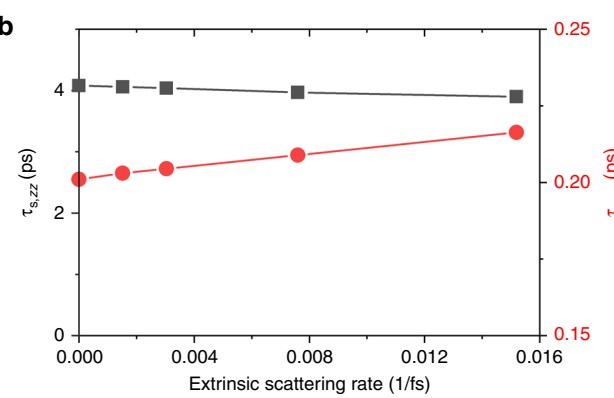

**Fig. 6 Dependence of MoS$_2$ spin relaxation on temperature and extrinsic scattering.** For conduction electrons in MoS$_2$, **a** variation of in-plane spin relaxation time ($\tau_{s,xx}$) with temperature, and **b** comparison of in-plane ($\tau_{s,xx}$) and out-of-plane ($\tau_{s,zz}$) spin lifetimes as a function of extrinsic scattering rates at 300 K.

**Systems without inversion symmetry: in-plane $\tau_s$ of MoS$_2$.** In all cases, the spin–phonon relaxation time decreases with increasing temperature, approximately proportional to the momentum relaxation time $\tau_m$. This is expected because both scattering rates, $\tau_s^{-1}$ and $\tau_m^{-1}$, are proportional to phonon occupation factors which increase with temperature. The intrinsic in-plane spin relaxation time ($\tau_{s,xx}$) in MoS$_2$ also shows the same trend with temperature (Fig. 6a), but exhibits a fundamental

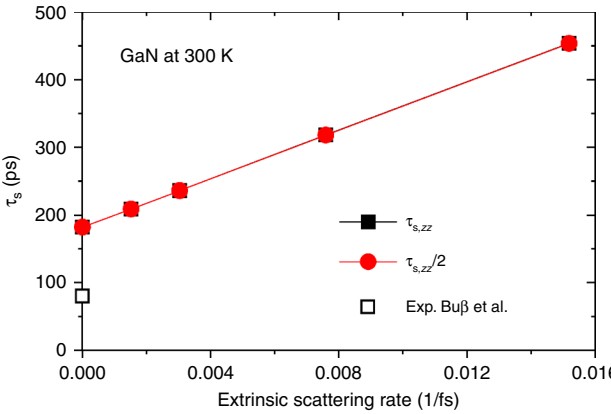

**Fig. 7 Dependence of GaN spin relaxation on extrinsic scattering.** In-plane ($\tau_{s,xx}$) and out-of-plane ($\tau_{s,zz}$) spin lifetime of GaN as a function of extrinsic scattering rates at 300 K. The experimental $\tau_{s,zz}$ at 298 K is from ref. [69].

difference from the previous cases when considering additional extrinsic scattering.

Specifically, Fig. 6b shows the dependence of spin relaxation times in $MoS_2$ conduction electrons as a function of extrinsic scattering rates, which enter Eq. (3) through an additional contribution to the smearing width $\gamma$ of the energy-conserving $\delta$-functions (in addition to the intrinsic electron–phonon contributions computed from first-principles). This additional smearing physically corresponds to the reduced lifetime and increased broadening of the electronic states in the material due to scattering against defects, impurities etc[37]. Importantly, the in-plane spin relaxation time $\tau_{s,xx}$ increases linearly with extrinsic scattering rate, or inversely with extrinsic scattering time, which is a hallmark of the DP mechanism of spin relaxation[48].

Note that this inverse relation competes with the phonon occupation factors in determining the overall temperature dependence of spin relaxation time. At higher temperature, increased phonon occupation factors lower the intrinsic relaxation times of both carrier and spin, as stated above. The lowered intrinsic carrier relaxation time increases the finite smearing width in Eq. (3), which contributes towards increasing the spin relaxation time within the DP mechanism (inverse relation). However, the direct contribution of phonon occupation factors in the spin relaxation rate in Eq. (3) overwhelms this secondary change and results in a net decrease of spin relaxation time, consistent with all calculations above and experiments[49,50].

In contrast with the in-plane case, the out-of-plane spin relaxation $\tau_{s,zz}$ is mostly insensitive to the extrinsic scattering rate (and broadening $\gamma$), as all previous spin relaxation results in Kramers-degenerate materials discussed above (e.g. for graphene in Fig. 2d). Note that $\tau_{s,xx}$ is also overall much shorter than $\tau_{s,zz}$, because the strong internal magnetic field in TMDs stabilizes spins in the $z$ direction as discussed above. Large anisotropy in spin lifetimes due to a similar spin-valley locking effect has been theoretically predicted[51] and experimentally measured[52] previously in graphene–TMD interfaces as well.

**Systems without inversion symmetry: GaN.** Finally, we show spin relaxation in GaN as an archetypal example of the DP mechanism. Fig. 7 shows that both in-plane ($\tau_{s,xx}$) and out-of-plane ($\tau_{s,zz}$) spin lifetime of GaN are proportional to extrinsic scattering rates, or inversely proportional to extrinsic scattering time. Most importantly, the ratio between $\tau_{s,zz}$ and $\tau_{s,xx}$ is exactly 1/2 for this material, which is an additional feature of the conventional DP mechanism[1]. Note that, in contrast, the 2D TMDs

are more complex due to strong SOC splitting and anisotropy, did not exhibit this 1/2 ratio, and exhibited the extrinsic scattering dependence only for in-plane spin relaxation. Overall, these results indicate that the general density-matrix formalism presented here elegantly captures the characteristic DP and EY mechanism limits, as well as complex cases that do not fit these limits, all on the same footing in a unified framework.

## Discussion

In summary, we have demonstrated an accurate and universal first-principles method for predicting spin relaxation time of arbitrary materials, regardless of electronic structure, strength of spin-mixing and crystal symmetry (especially with/without inversion symmetry). Our work goes far beyond previous first-principles techniques based on a specialized Fermi's golden rule with spin–flip transitions and provides a pathway to an intuitive understanding of spin relaxation with arbitrary spin-mixing. In TMD monolayer materials, we clarify the roles of intravalley and intervalley processes, which are additionally resolved by phonon modes, in electron and hole–spin relaxation. We predict long-lived spin polarization from resident carriers of $MoS_2$ and $MoSe_2$ and show their strong sensitivity of electron spin relaxation to in-plane magnetic fields.

The predictive power of first-principles calculations is crucial for providing fundamental understanding of spin relaxation in new materials. The same technique can be applied to predict spin relaxation in realistic materials with or without defects useful for quantum technologies, wherever spin relaxation is dominated by electron–phonon scattering. We have already considered the general impact of disorder and electron-impurity scattering on spin–phonon relaxation through carrier broadening, but impurities can contribute an additional channel for spin relaxation, especially in the Kramers-degenerate case and at lower temperatures[18]. The extension of this technique to directly predict electron-impurity scattering for specific defects is relatively straightforward using supercell calculations, but computationally more demanding, while predicting the impact of electron–hole interaction[53–55] and electron–electron scattering[56,57] is additionally challenging. Finally, a robust understanding of ultrafast experiments may require simulation of real-time dynamics to capture initial state effects, probe wavelength effects and beyond-single-exponential decay dynamics, which is a natural next step within the general Lindbladian density-matrix formalism presented here.

## Methods

**Computational details**. All simulations are performed by the open-source plane-wave code - JDFTx[58] using pseudopotential method, except that the Born effective charges and dielectric constants are obtained from open-source code QuantumESPRESSO[59]. We firstly carry out electron structure, phonon and electron–phonon matrix element calculations in DFT using Perdew–Burke–Ernzerhof exchange-correlation functional[60] with relatively coarse **k** and **q** meshes. The phonon calculations are done using the supercell method. We have used supercells of size $7 \times 7 \times 7$, $4 \times 4 \times 4$, $6 \times 6 \times 1$, $6 \times 6 \times 1$, $6 \times 6 \times 1$, $4 \times 4 \times 4$ for silicon, BCC iron, graphene, monolayer $MoS_2$, monolayer $MoSe_2$ and GaN, respectively, which have shown reasonable convergence for each system (<20% error bar in the final spin relaxation estimates). SOC is included through the use of the fully relativistic pseudopotentials[61]. For monolayer $MoS_2$ and $MoSe_2$, the Coulomb truncation technique is employed to accelerate convergence with vacuum sizes[62].

We then transform all quantities from plane wave to maximally localized Wannier function basis[63] and interpolate them[24,25,64,65] to substantially finer **k** and **q** meshes (with $>3 \times 10^5$ total points) for lifetime calculations. Statistical errors of lifetime computed using different random samplings of $k$-points are found to be negligible (<1%). This Wannier interpolation approach fully accounts for polar terms in the electron–phonon matrix elements and phonon dispersion relations using the approaches of Verdi and Giustino[66] and Sohier et al.[67] for the 3D and 2D systems.

## Data availability

All relevant data are available from the authors upon request.

## Code availability

First-principles methodologies available through open-source software, JDFTx[58], and post-processing codes available from authors upon request.

58. Sundararaman, R. et al. Jdftx: software for joint density-functional theory. *SoftwareX* **6**, 278–284 (2017).

59. Giannozzi, P. et al. Quantum espresso: a modular and open-source software project for quantum simulations of materials. *J. Condens. Matter Phys.* **21**, 395502 (2009).

60. Perdew, J. P., Burke, K. & Ernzerhof, M. Generalized gradient approximation made simple. *Phys. Rev. Lett.* **77**, 3865 (1996).

61. Van Setten, M. et al. The pseudodojo: training and grading a 85 element optimized norm-conserving pseudopotential table. *Comput. Phys. Commun.* **226**, 39–54 (2018).

62. Sundararaman, R. & Arias, T. Regularization of the Coulomb singularity in exact exchange by Wigner–Seitz truncated interactions: towards chemical accuracy in nontrivial systems. *Phys. Rev. B* **87**, 165122 (2013).

63. Marzari, N. & Vanderbilt, D. Maximally localized generalized wannier functions for composite energy bands. *Phys. Rev. B* **56**, 12847 (1997).

64. Giustino, F., Cohen, M. L. & Louie, S. G. Electron-phonon interaction using wannier functions. *Phys. Rev. B* **76**, 165108 (2007).

65. Brown, A. M. et al. Experimental and ab initio ultrafast carrier dynamics in plasmonic nanoparticles. *Phys. Rev. Lett.* **118**, 087401 (2017).

66. Verdi, C. & Giustino, F. Fröhlich electron-phonon vertex from first principles. *Phys. Rev. Lett.* **115**, 176401 (2015).

67. Sohier, T., Gibertini, M., Calandra, M., Mauri, F. & Marzari, N. Breakdown of optical phonons' splitting in two-dimensional materials. *Nano Lett.* **17**, 3758–3763 (2017).

68. Towns, J. et al. Xsede: accelerating scientific discovery. *Comput. Sci. Eng.* **16**, 62–74 (2014).

69. Buß, J., Rudolph, J., Natali, F., Semond, F. & Hägele, D. Temperature dependence of electron spin relaxation in bulk gan. *Phys. Rev. B* **81**, 155216 (2010).

## Acknowledgements

We thank Ming-wei Wu, Oscar Restrepo, and Wolfgang Windl for helpful discussions. This work is supported by National Science Foundation under Grant Nos. DMR-1760260, and startup funding from the Department of Materials Science and Engineering at Rensselaer Polytechnic Institute. This research used resources of the Center for Functional Nanomaterials, which is a US DOE Office of Science Facility, and the Scientific Data and Computing center, a component of the Computational Science Initiative, at Brookhaven National Laboratory under Contract No. DE-SC0012704, the lux supercomputer at UC Santa Cruz, funded by NSF MRI Grant AST 1828315, the National Energy Research Scientific Computing Center (NERSC), a U.S. Department of Energy Office of Science User Facility operated under Contract No. DE-AC02-05CH11231, the Extreme Science and Engineering Discovery Environment (XSEDE), which is supported by National Science Foundation Grant No. ACI-1548562[68], and resources at the Center for Computational Innovations at Rensselaer Polytechnic Institute.

## Author contributions

J.X. and A.H. implemented the codes and performed the major part of ab initio calculations; S.K. and F.W. contributed to part of the calculations and data analysis; J.X., A.H., R.S. and Y.P. wrote the manuscript with contributions from all authors. Y.P. and R.S. designed and supervised all aspects of the project.

## Competing interests

The authors declare no competing interests.
