## [Peer Review File · Nature Communications]

Reviewers' comments:

Reviewer #1 (Remarks to the Author):

In this manuscript, Xu and co-workers present a numerical approach to simulate spin dynamics in materials in presence of spin orbit coupling and electron-phonon interaction. They apply their method to Silicon, Iron and transition metal dichalcogenides materials (TMD), and compare favorably their temperature dependent spin lifetime with available experimental data.

Overall, this study is of high quality and indeed brings a very interesting perspective on phonon-mediated spin transport and spin relaxation phenomena. However, there are some essential points that must be clarified and some extra calculations which would be necessary to reach a fully consistent and useful contribution to the community.

The approach developed by the authors does not allow one to include static disorder which will break translational invariance and play a key role at low temperatures. Additionally, the Mathiessen rule is expected to break down for low dimensional materials such as few layers TMD, and static defects such as vacancies are likely to also generate inelastic disorder very specific to the breaking of crystal symmetries. To which extent this could affect some of the conclusion of the paper is difficult to say, but cannot be disregarded. Actually for TMD, the authors should add an important calculation which is to show the result of the in-place spin lifetime using the same electron-phonon driven scattering times.

I also have a couple of true concerns about the methodology as well as the results presented. The authors state that their method allows the consideration of Elliott-Yafet (EY) and D'yakonov-Perel' (DP) spin relaxation on equal footing, yet all of their results appear to show only EY spin relaxation. I do not doubt that their approach might also capture DP spin relaxation behavior, but the main message of this paper seems to be the universality and flexibility of their method. Without some results backing up their claims of generality, this paper loses some of its impact, in my opinion. For this reason, I would recommend they also show results for a system where DP is the dominant spin relaxation mechanism. The conventional DP mechanism actually presents a few universalities such as the 1/2 ratio between out-of-plane and in-plane spin lifetime, or the inverse scaling of the spin lifetime with momentum scattering. Since the scaling seems here different, the ratio presents some particular interest, and the authors should show what they obtain for TMDs.

Next, I am particularly concerned about the assumption that the spin density decays exponentially, as given in Eq. (2). This assumption is fine for EY and DP, but as soon as one leaves the motional narrowing regime the DP mechanism breaks down and the spin density is characterized by a relaxation plus a coherent precession. This occurs when spin-orbit coupling is strong, i.e., exactly the scenario encountered in TMD. Can the authors verify that the time-dependent decay of the spin density is consistent with the spin relaxation time extracted with the assumption of purely exponential decay? I would feel more comfortable about their method and results if they could do this for at least one data point. Looking at the full dynamics can also bring much more information, and may even shed light on what is actually being measured, given the coincidence between the simulations and the experimental results. I think that in-plane vs. out-of-plane spin dynamics in TMD may also show quite different behaviors.

Finally, it would be truly interesting to see what result is obtained by the proposed methodology for the clean case of graphene. This should be straightforward for the authors given the simplicity of the material, and would allow a direct comparison with analytical results or results which have little temperature-dependence below 200K (see A.W. Cummings and S. Roche, Phys. Rev. Lett. 116 (8), 086602 (2016) and D. V. Tuan et al., Scientific Reports 6, 21046 (2016)). It is indeed an open

question to know if substrate effects, pure dephasing or (maybe) the electron-phonon coupling (discussed in Fratini et al, Phys. Rev. B 88, 115426 (2013)) are the ultimate sources of spin decoherence in ultraclean graphene.

Adding some data in that respect would make the manuscript of genuine relevance for the field of spin transport.

Reviewer #2 (Remarks to the Author):

This paper presents a comprehensive theoretical approach to calculate spin relaxation times fully from first principles. The approach is formally based on density-matrix theory within the Lindblad formalism, where the spins relax due to a combination of electron-phonon scattering and spin-orbit coupling. In this way, the important Elliot-Yafet and Dyakonov-Perel mechanisms are both covered, without any bias towards one or the other. The input to the calculations consists of the electronic band structure (calculated with DFT, including spin-orbit coupling) and the electron-phonon scattering elements. All of these are obtained using standard electronic structure methods. The final step is then to combine these ingredients in a Fermi's-Golden-Rule-like formula for the spin-phonon relaxation time. The formalism is carefully derived in the supplemental information.

The approach is then applied to three different materials: silicon, iron, and the transition metal dichalcogenides MoS₂ and MoSe₂. Both electron and hole spin relaxation times are calculated as a function of temperature, and compared with available experimental data. The agreement is very good, which suggests that the approach presented here is powerful and versatile. Moreover, the results are carefully analyzed in terms of underlying mechanisms such as valley relaxation and magnetic-field dependence. Several new insights are obtained, which will be of interest to experts in the field.

The paper is scientifically sound, well written and very clear. The research is in an area of topical interest (the material science of spintronics). The approach to calculating spin lifetimes appears superior to what has been previously done in the literature. This work is likely to have broad impact in the field. I recommend publication.

I have only a few relatively minor comments.

1. I would suggest to slightly alter the title of the paper. The phrase "in disparate materials" is both too vague and too narrow. "Disparate" is usually understood to mean two utterly incompatible things or concepts, to the extent that they cannot even be compared with each other. This, I believe, is not what the authors mean here (they just mean very different materials). I think that it would be better to drop this phrase completely, and just call the paper "Spin-phonon relaxation from a universal ab initio density-matrix approach".

2. On page 2 of the main paper (left column, bottom line), and also in equation (1) in the supplemental material (middle formula), the quantity G is defined to contain a square root of a delta function. This is formally ill-defined: square roots of delta functions have no formal meaning. I understand that in the final formula (when the G 's are properly combined) all works out just fine, but nevertheless this is problematic. I would suggest to redefine the quantities to avoid square roots of delta functions. I checked the paper by Rosati, Dolcini and Rossi, and I could not find a square root of a delta function there (they used a Gaussian regularization of the delta function).

3. In the section "Outlook", first paragraph, line 14, it should be MoS₂ and MoSe₂.

Reply to reviewer 1:

“In this manuscript, Xu and co-workers present a numerical approach to simulate spin dynamics in materials in presence of spin orbit coupling and electron-phonon interaction. They apply their method to Silicon, Iron and transition metal dichalcogenides materials (TMD), and compare favorably their temperature dependent spin lifetime with available experimental data.

Overall, this study is of high quality and indeed brings a very interesting perspective on phonon-mediated spin transport and spin relaxation phenomena.

However, there are some essential points that must be clarified and some extra calculations which would be necessary to reach a fully consistent and useful contribution to the community.”

We greatly appreciate the reviewer’s overall positive response and are thankful for the constructive feedback and suggestions which have allowed us to substantially strengthen our manuscript.

“The approach developed by the authors does not allow one to include static disorder which will break translational invariance and play a key role at low temperatures. Additionally, the Mathiessen rule is expected to break down for low dimensional materials such as few layers TMD, and static defects such as vacancies are likely to also generate inelastic disorder very specific to the breaking of crystal symmetries. To which extent this could affect some of the conclusion of the paper is difficult to say, but cannot be disregarded.”

We thank the reviewer for this important question, and completely agree that static disorder and defects can be important for spin lifetime, especially at low temperature. In fact, our approach can naturally take into account the effect of defects and static disorder from first principles by including defect explicitly within supercell calculations. Previous first-principles calculations of EY-only spin relaxation [O. D. Restrepo and W. Windl, *Phys. Rev. Lett.* 109, 166604, (2012)] have estimated defect contributions without explicit supercells using impurity potentials, but that is not a strictly first-principles approach that generalizes to all materials.

We have not included spin scattering due to explicit defects in supercells in this first manuscript highlighting the approach for two reasons. First, the computational cost of explicit supercell calculations would be significantly higher and requires further optimization and parallelization of the code, which we are carrying out in parallel. Second, the scattering effects due to defects would require considering specific defects in specific host materials in the first-principles calculations and would not help showcase the generality of our approach across several materials. Consequently, we chose to limit our scope to *intrinsic* spin scattering rate, where we can compare to several materials with varying symmetry classes and electronic structures without trying to match specific defect contributions when comparing to experiment.

In the revised manuscript, we have also mentioned in the outlook that explicit defect scattering can be considered straightforwardly using the approach established here then applied to supercells, but at additional computational cost. Finally, as discussed below, we have also shown the impact of *extrinsic* scattering due to static disorder and defects (treated generally through their corresponding carrier linewidth contribution) on spin relaxation in the DP mechanism. Related discussions and calculations have been added to subsections of “in-plane τ_s of MoS₂”, “Spin relaxation in GaN” and outlook on page 8 and 9.

“Actually for TMD, the authors should add an important calculation which is to show the result of the in-plane spin lifetime using the same electron-phonon driven scattering times.”

We thank the reviewer for this useful suggestion: we have added new calculations of in-plane and out-of-plane spin lifetime for MoS₂ in the revised manuscript. Figure 1 below (Figure 6 in the main text) shows that $\tau_{s,xx}$ (in-plane lifetime) is about one order of magnitude smaller than $\tau_{s,zz}$ (out-of-plane lifetime). More importantly, they have distinct dependence with the extrinsic scattering rates, such as impurity scatterings, which are now included in our revised calculations as extra constant smearing parameters of the energy conserving delta function in the generalized rate equation (Eq. 3 of the main text).

Specifically, in-plane $\tau_{s,xx}$ increases with increasing extrinsic scattering rate, which is expected for spin relaxation due to DP mechanism. On the other hand, the out-of-plane $\tau_{s,zz}$ is less sensitive to the extrinsic scattering rate.

This clearly indicates the qualitative difference in spin relaxation mechanisms between in-plane and out-of-plane directions, and confirms previous work showing similar effects using model Hamiltonians, such as in L. Wang et al, PRB 89, 115302(2014).

We have included these new results as Figure 6 in the main text, along with a corresponding discussion of the above results. (See subsection “In-plane $\tau_{s,xx}$ of MoS₂” on page 8.)

Figure 1. In-plane ($\tau_{s,xx}$) and out-of-plane ($\tau_{s,zz}$) spin lifetime of MoS₂ as a function of extrinsic scattering rates at 300 K.

“I also have a couple of true concerns about the methodology as well as the results presented. The authors state that their method allows the consideration of Elliott-Yafet (EY) and D’yakonov-Perel’ (DP) spin relaxation on equal footing, yet all of their results appear to show only EY spin relaxation. I do not doubt that their approach might also capture DP spin relaxation behavior, but the main message of this paper seems to be the universality and flexibility of their method. Without some results backing up their claims of generality, this paper loses some of its impact, in my opinion. For this reason, I would recommend they also show results for a system where DP is the dominant spin relaxation mechanism. The conventional DP mechanism actually presents a few universalities such as the 1/2 ratio between out-of-plane and in-plane spin lifetime, or the inverse scaling of the spin lifetime with momentum scattering. Since the scaling seems here different, the ratio presents some particular interest, and the authors should show what they obtain for TMDs.”

The reviewer brings up an extremely important point about specifically showcasing the DP mechanism, and we are extremely grateful for this suggestion because it allows us to most clearly showcase the generality of our first-principles approach. We have added a number of calculations to the revised manuscript to address this critical point such as in-plane spin relaxation of TMDs and GaN.

First, as discussed above in response to the previous point, we now compare in-plane and out-of-plane spin scattering times as a function of extrinsic scattering rate in TMDs such as MoS₂. We find the in-plane spin life time to be proportional to the extrinsic scattering rates (i.e. inversely proportional to carrier lifetime due to defects) in Figure 1, as expected for the DP mechanism. We do not see a ½ ratio between out-of-plane and in-plane spin lifetimes for TMDs. In fact, the ½ ratio is derived for two-dimensional III-V quantum wells and heterostructures with broken inversion symmetry as discussed in *Rev. Mod. Phys.*, 76, 323, 2016, and may not work exactly for TMDs, partly because the out-of-plane spin is linked intricately with the valley degree of freedom, and the internal effective magnetic field is nearly along the same direction z with it.

Second, we showcase the ½ ratio in other systems that are archetypal examples of the DP mechanism. Specifically, we show for GaN in Figure 2 that both the in-plane and out-of-plane spin relaxation time are directly proportional to the extrinsic scattering rate (inverse of carrier lifetime due to defects, impurities etc.), and exhibit the ½ ratio exactly. We have included Figure 2 as Figure 7 in the main text of the revised manuscript, accompanied by the above discussion in the subsection “spin relaxation in GaN” on pages 8-9. This clearly establishes the capability of our method to elegantly capture the DP mechanism in the same general framework, and thank the reviewer for the comments that led to its inclusion.

Figure 2: In-plane ($\tau_{s,xx}$) and out-of-plane ($\tau_{s,zz}$) spin lifetime as a function of extrinsic scattering rate for GaN, clearly showcasing the ½ ratio and inverse relation with carrier lifetime (inverse of extrinsic scattering rate) characteristic of the DP mechanism at 300 K compared with the experimental value (at 298 K) from *Physical Review B* 81, 155216 (2010).

“Next, I am particularly concerned about the assumption that the spin density decays exponentially, as given in Eq. (2). This assumption is fine for EY and DP, but as soon as one leaves the motional narrowing regime the DP mechanism breaks down and the spin density is characterized by a relaxation plus a coherent precession. This occurs when spin-orbit coupling is strong, i.e., exactly the scenario encountered in TMD. Can the authors verify that the time-dependent decay of the spin density is consistent with the spin relaxation time extracted with the assumption of purely exponential decay? I would feel more comfortable about their method and results if they could do this for at least one data point. Looking at the full dynamics can also bring

much more information, and may even shed light on what is actually being measured, given the coincidence between the simulations and the experimental results. I think that in-plane vs. out-of-plane spin dynamics in TMD may also show quite different behaviors.”

We completely agree with the reviewer that real-time dynamics using a first-principles electron-phonon formalism will be extremely useful. We are working precisely in that direction for future work, but significant computational developments are required to achieve those calculations. Specifically, an explicit time evolution using ab initio matrix elements requires approximately *three to four orders of magnitude* more computation than the decay rate calculations presented here. Algorithmic work that takes advantage of e-ph matrix element sparsity and then parallelizes efficiently over them is underway in our group in order to achieve such calculations.

In the meantime, we tested using simpler (not *ab initio*) models that exhibit both precession and relaxation, and find that the spin density typically decays as $S(t) = S_0 \exp(-t/\tau) \cos(\omega t)$. For such relaxation, the rate of change of the spin expectation value at the initial time is $S'(0) = -S_0/\tau$, which agrees with the rate expression based on a pure exponential decay. Therefore, we expect our rate expression to give the correct spin relaxation rate even for cases that include precession. For instance, we used the same model Hamiltonian as in L. Wang et al, PRB 89, 115302(2014), by our generalized rate formula, we obtained in-plane spin relaxation time of MoS₂ at 300 K for carrier concentration $7 \times 10^{12} \text{ cm}^{-2}$, 0.9 ps, similar to the value obtained from the fitting of the real-time dynamics results with $\exp(-t/\tau) \cos(\omega t)$, about 0.6 ps for τ (see Figure 3). We thank the reviewer once again for raising this important concern: we have briefly discussed the validity of our predictions in cases with precession based on the above argument below Eq.2 on page 3 of the revised manuscript, and outlined the importance of future real-time dynamics simulations for strong SOC precession cases as well as direct experimental signature comparison in the outlook section.

Figure 3: Real-time evolution of $|S_x(t)|$ of a model system of MoS₂ at 300 K by the same density matrix master equation as in PRB 89, 115302(2014) but considering only the electron-phonon scattering and its fit with a single oscillation part and a single decay part ($\exp(-t/\tau) \cos(\omega t)$). The obtained spin relaxation time from the fit is about 0.6 ps in agreement with the value 0.9 ps, computed by our proposed generalized rate formula (Eq. 3 in the main text).

“Finally, it would be truly interesting to see what result is obtained by the proposed methodology for the clean case of graphene. This should be straightforward for the authors given the simplicity of the material, and would allow a direct comparison with analytical results or results which have little temperature-dependence below 200K (see A.W. Cummings and S. Roche, *Phys. Rev. Lett.* 116 (8), 086602 (2016) and D. V. Tuan et al., *Scientific Reports* 6, 21046 (2016)). It is indeed an open question to know if substrate effects, pure dephasing or (maybe) the electron-phonon coupling (discussed in Fratini et al, *Phys. Rev. B* 88, 115426 (2013)) are the ultimate sources of spin decoherence in ultraclean graphene.

Adding some data in that respect would make the manuscript of genuine relevance for the field of spin transport.”

Following the reviewer’s suggestion, we applied our general *ab initio* technique to evaluate electron-phonon (intrinsic) spin relaxation in graphene as a function of Fermi level position, as shown in Figure 3. We find overall similar magnitudes of electron-phonon spin relaxation time and qualitative similarities in the dependence with Fermi level position as previous model Hamiltonian predictions in [*PRB* 88, 115426 (2013)] and [*PRB* 95, 195402 (2017)]. Note however that our *ab initio* predictions include all phonon modes with explicit phonon dispersion relations and e-ph matrix elements in the rate expression, compared to flexural mode only with low q expansions of the dispersion and model matrix elements in previous work. Indeed, we find that the flexural mode dominates, but the contribution due to in-plane acoustic phonons is not negligible. However, the magnitude of spin lifetimes is larger than experiment, indicating that spin relaxation due to phonons is likely not the dominant mechanism. We also find that the ratio between in-plane and out-of-plane spin relaxation times ranges from 0.5 - 0.7, in excellent agreement with experimental measurements. We have included Figure 3 as Figure 2 in the main text, along with the above discussion. See subsection “Graphene” on pages 4-5. The new results further showcase the general utility of our approach, and we thank the reviewer for suggesting their inclusion.

Figure 4. Intrinsic electron-phonon spin relaxation time for free-standing graphene (a,b) at room temperature for various Fermi energies, and (c) at $E_F = 0.1$ eV as a function of temperature. Panel (b) shows out-of-plane spin relaxation by cumulative phonon mode contributions starting with optical (O), adding in-plane acoustic (A), and then including the flexural mode as well, indicating the dominance of acoustic and flexural modes for spin relaxation. Both (a) and (c) show an anisotropy between in-plane and out-of-plane lifetimes with a ratio varying in the 0.5 - 0.7 range.

Reply to Reviewer: 2

“This paper presents a comprehensive theoretical approach to calculate spin relaxation times fully from first principles. The approach is formally based on density-matrix theory within the Lindblad formalism, where the spins relax due to a combination of electron-phonon scattering and spin-orbit coupling. In this way, the important Elliot-Yafet and Dyakonov-Perel mechanisms are both covered, without any bias towards one or the other. The input to the calculations consists of the electronic band structure (calculated with DFT, including spin-orbit coupling) and the electron-phonon scattering elements. All of these are obtained using standard electronic structure methods. The final step is then to combine these ingredients in a Fermi's-Golden-Rule-like formula for the spin-phonon relaxation time. The formalism is carefully derived in the supplemental information.

The approach is then applied to three different materials: silicon, iron, and the transition metal dichalcogenides MoS₂ and MoSe₂. Both electron and hole spin relaxation times are calculated as a function of temperature, and compared with available experimental data. The agreement is very good, which suggests that the approach presented here is powerful and versatile. Moreover, the results are carefully analyzed in terms of underlying mechanisms such as valley relaxation and magnetic-field dependence. Several new insights are obtained, which will be of interest to experts in the field.

The paper is scientifically sound, well written and very clear. The research is in an area of topical interest (the material science of spintronics). The approach to calculating spin lifetimes appears superior to what has been previously done in the literature. This work is likely to have broad impact in the field. I recommend publication.”

We thank the reviewer for the strongly positive comments on our work, and agree completely with the synopsis.

“I have only a few relatively minor comments.

1. I would suggest to slightly alter the title of the paper. The phrase "in disparate materials" is both too vague and too narrow. "Disparate" is usually understood to mean two utterly incompatible things or concepts, to the extent that they cannot even be compared with each other. This, I believe, is not what the authors mean here (they just mean very different materials). I think that it would be better to drop this phrase completely, and just call the paper "Spin-phonon relaxation from a universal ab initio density-matrix approach".

We thank the reviewer for the suggestion and have updated the title accordingly.

“2. On page 2 of the main paper (left column, bottom line), and also in equation (1) in the supplemental material (middle formula), the quantity G is defined to contain a square root of a delta function. This is formally ill-defined: square roots of delta functions have no formal meaning. I understand that in the final formula (when the G 's are properly combined) all works out just fine, but nevertheless this is problematic. I would suggest to redefine the quantities to avoid square roots of delta functions. I checked the paper by Rosati, Dolcini and Rossi, and I could not find a square root of a delta function there (they used a Gaussian regularization of the delta function).”

We thank the reviewer for pointing this out. We indeed used a Gaussian instead of a delta function in the practical implementation similar to the original paper by Rosati et al., and have clarified this in our explanation of Eq. 1 in the second paragraph on page 3 of the revised manuscript.

“3. In the section "Outlook", first paragraph, line 14, it should be MoS2 and MoSe2.”

We have fixed this typographical error, and thank the reviewer for pointing it out.

Once again, we thank the reviewers for their time and constructive feedback, all of which have substantially strengthened our revised manuscript that should now be suitable for publication in *Nature Communications*.

Reviewers' comments:

Reviewer #1 (Remarks to the Author):

The revised version of the manuscript and new results make this work of very high interest. I think that with the additional (final!) suggestions I am making below, such a paper will become a milestone of the field of spin relaxation. So I encourage the authors to add the following

1- For the case of Graphene, the simulations show two features which apparently seem to contradict each other. Indeed, the ratio between out-of-plane spin lifetime and in-plane is about $\frac{1}{2}$, in full coincidence with the Dyakonov Perel regime for spin relaxation. The increase of the spin lifetime with decreasing Fermi level is also similar to what was found by Ertler et al Phys. Rev. B 80, 041405(R) (2009), using Monte carlo simulations of electron transport in presence of a static random spin-orbit coupling fields.

However simultaneously, the spin lifetime computed here (Figure 2(c)) decays with increasing temperature, which would at first sight contradict the Dyakonov-Perel mechanism following the work mentioned above. However I think the main difference here is that the author's simulation frame entangles vibrational disorder with fluctuations of spin-orbit coupling terms, so that the higher the temperature, the stronger the fluctuations of the SOC field and thus the faster the spin relaxation. This to me seems a plausible interpretation, which should be of main interest!

I thus encourage the authors to comment on that, but also to try to extract the inelastic scattering time (τ_{in}) versus temperature from their simulations so as to show a new curve of how the spin lifetime (τ_s) scales with inelastic scattering time?

If the scenario is correct, the authors will find that $\tau_s = 1/\tau_{in}$, therefore manifesting the universality of Dyakonov-Perel and the absence of contradiction with works such as Ertler et al Phys. Rev. B 80, 041405(R) (2009).

1- For the case of MoS2, the results and conclusion are also excellent...the authors find that the out-of-plane spin lifetime can be about 10 times the in-plane spin lifetimes as a results of internal SOC field. Actually the term at the origin of such effect is the Valley Zeeman coupling which could certainly be defined by some reparametrized tight-binding model of MoS2. The point is that such anisotropy corresponds perfectly to what has been widely discussed theoretically for graphene/TMD (transition metal dichalcogenide). In this case the TMD imprints SOC on graphene and a similar spin transport anisotropy was predicted (Physical Review Letters 119 (20), 206601 (2017)) and measured experimentally (Nature Physics 14 (3), 303-308 (2018)). The observation of such huge anisotropy however requires disorder so as to enter a similar "Dyakonov-Perel regime", although not dominated anymore by a Rashba-type of SOC fields.

I am strongly recommending to the authors to make such connection more explicit, since it gives further validation of their simulations and show the universalities of spin transport anisotropy with respect to SOC effects. Again the temperature dependence found in the simulations of Xu et al could be seen as contradicting the analogy, but the same argument as discussed for graphene is likely to apply, mainly that the change in temperature is concomitant with a varying strength of SOC fields active in the ab-initio calculations.

I hope the authors will admit that if they revise the manuscript adding the ingredients above, their contribution will have an even larger impact in the field of spin transport in nanomaterials.

Reviewer #2 (Remarks to the Author):

The authors have submitted a revised manuscript, in which they have addressed the comments and concerns of the previous referee reports, and included a sizable amount of new results. The detailed

responses to the points raised in the previous reviews are convincing. The paper is now much stronger, and definitely suitable for publication in Nature Communications.

We are extremely grateful for the reviewers' constructive comments and your consideration of our manuscript. We did additional revisions based on the reviewer's comments aimed at improving the impact and broader interest of our manuscript. We believe our work should now be suitable for publication in *Nature Communications*.

Below, we repeat the reviewer's comments in *black italics*, and present our responses point-by-point in blue color.

Reply to reviewer 1:

"The revised version of the manuscript and new results make this work of very high interest. I think that with the additional (final!) suggestions I am making below, such a paper will become a milestone of the field of spin relaxation. So I encourage the authors to add the following"

We greatly appreciate the encouraging comments from the reviewer! These final comments have been extremely helpful for us to significantly clarify certain central messages of the manuscript, especially with regard to identifying the Dyakonov-Perel (DP) regime.

"1- For the case of Graphene, the simulations show two features which apparently seem to contradict each other. Indeed, the ratio between out-of-plane spin lifetime and in-plane is about $\frac{1}{2}$, in full coincidence with the Dyakonov Perel regime for spin relaxation. The increase of the spin lifetime with decreasing Fermi level is also similar to what was found by Ertler et al Phys. Rev. B 80, 041405(R) (2009), using Monte carlo simulations of electron transport in presence of a static random spin-orbit coupling fields.

However simultaneously, the spin lifetime computed here (Figure 2(c)) decays with increasing temperature, which would at first sight contradict the Dyakonov-Perel mechanism

following the work mentioned above. However I think the main difference here is that the author's simulation frame entangles vibrational disorder with fluctuations of spin-orbit coupling terms, so that the higher the temperature, the stronger the fluctuations of the SOC field and thus the faster the spin relaxation. This to me seems a plausible interpretation, which should be of main interest!

I thus encourage the authors to comment on that, but also to try to extract the inelastic scattering time (τ_{in}) versus temperature from their simulations so as to show a new curve of how the spin lifetime (τ_s) scales with inelastic scattering time?

If the scenario is correct, the authors will find that $\tau_s = 1/\tau_{in}$, therefore manifesting the universality of Dyakonov-Perel and the absence of contradiction with works such as Ertler et al Phys. Rev. B 80, 041405(R) (2009)."

The reviewer brings attention to an interesting point about the nature of spin relaxation in graphene. The nearly $\frac{1}{2}$ ratio between in-plane and out-of-plane spin relaxation could be seen as suggesting a DP mechanism. In the revised manuscript, we include an additional calculation of the spin relaxation time in graphene as a function of extrinsic scattering rates to clarify this point (Fig. 1 below and new panel Fig. 2(d) in the manuscript), as we describe below.

First, note that we did not break the inversion symmetry of graphene, e.g. using substrates or an external electric field, and we therefore do not expect its spin relaxation to be in the DP regime. Here, we focus on intrinsic spin relaxation time of free-standing graphene, with substrate and external field effects beyond the scope of the current work. In contrast, the paper indicated by the reviewer studied graphene on SiO_2 substrates under an external electric field, which breaks inversion symmetry and causes a Bychkov-Rashba term in the Hamiltonian. Therefore, it is reasonable to expect the DP mechanism in that work, but not here.

Second, note that in the indicated paper, spin relaxation is mostly due to electron-impurity scattering, which is temperature-independent. While in our case, when we study temperature-dependence of spin lifetime, only electron-phonon scattering is considered. We would like to point out that even for systems with a DP mechanism, past experimental and theoretical work shows that spin relaxation time decreases with increasing temperature. For example, GaAs is a typical DP system, and it shows decreasing spin lifetime with increasing temperature in Refs: S. Krishnamurthy et al, Appl.Phys.Lett. 83, 1762, (2003); S. Oertel et al, Appl.Phys.Lett. 93, 132112 (2008), J. M. Kikkawa et al, 80, 4313, (1998); Phys. Rev. B 79, 125206 (2009).

Spin lifetime being inversely correlated with temperature including the DP case is a net result of different competing effects. At higher temperature, increased phonon occupation factors lower the intrinsic relaxation times of both carrier and spin. The lowered intrinsic carrier relaxation time that enters the finite smearing parameters in our rate formula (Eq. 3 of the main text) can contribute towards increasing the spin relaxation time within the DP mechanism (inverse relation), consistent with our theoretical results in Fig. 6(b) and 7 in the main text. However, the direct contribution of phonon occupation factors in the spin relaxation rate (according to our formula, Eq. 1, 3 and 5) could overwhelm this change. Moreover, we agree with the reviewer that, in many cases, the fluctuations of the SOC field (called internal magnetic field in our manuscript) are stronger at higher temperatures. This will make spin relaxation faster according to previous theoretical studies of DP systems (Žutić et al., Rev. Mod. Phys. 76, 323 (2004)). Although the fluctuations of the SOC field have been included in our theoretical framework, they are entangled with all other quantities. Therefore, it is unfortunately not obvious to conclude their effects on spin relaxation based on our method and further studies are required.

Overall, we conclude that when electron-phonon scattering processes dominate, the above effects together will result in a net decrease of spin relaxation time with increasing temperature in many cases, consistent with our theoretical results in Fig. 6(a) and previous theoretical and experimental work as mentioned above. We have included this discussion in the second paragraph on Page 9.

Most importantly, we observe the hallmark inverse relation between spin and momentum relaxation for systems with dominant DP mechanism as a function of extrinsic scattering rate, and not as a function of temperature. In the previous revision, we added these results for MoS₂ and GaN to showcase the DP mechanism. We have now included similar calculations for free-standing graphene, as shown below in Fig 1. It clearly shows that regardless of the Fermi level position and temperature, the spin relaxation time in free-standing graphene is mostly insensitive to the extrinsic scattering rate. This confirms that it is not the DP mechanism, and the nearly ½ ratio is likely coincidental.

We have included this figure as a new panel Fig. 2(d) and the above discussion at the end of the graphene section in the revised manuscript. We also added the momentum relaxation time to Fig. 2(c) in the revised manuscript, which decreases with increasing temperature as expected. These additions unambiguously clarify that free-standing graphene does not exhibit the DP mechanism, and we thank the reviewer for bringing it to our attention.

Fig. 1: Spin relaxation time in free-standing graphene is mostly insensitive to extrinsic scattering rate and does not show the proportionality characteristic of the DP mechanism.

“1- For the case of MoS₂, the results and conclusion are also excellent....the authors find that the out-of-plane spin lifetime can be about 10 times the in-plane spin lifetimes as a results of internal SOC field.

Actually the term at the origin of such effect is the Valley Zeeman coupling which could certainly be defined by some reparametrized tight-binding model of MoS₂. The point is that such anisotropy corresponds perfectly to what has been widely discussed theoretically for graphene/TMD (transition metal dichalcogenide).

In this case the TMD imprints SOC on graphene and a similar spin transport anisotropy was predicted (Physical Review Letters 119 (20), 206601 (2017)) and measured experimentally (Nature Physics 14 (3), 303-308 (2018)). The observation of such huge anisotropy however

requires disorder so as to enter a similar “Dyakonov-Perel regime”, although not dominated anymore by a Rashba-type of SOC fields.

I am strongly recommending to the authors to make such connection more explicit, since it gives further validation of their simulations and show the universalities of spin transport anisotropy with respect to SOC effects.”

We greatly appreciate the reviewer pointing out the connection with the past work and agree with the connection between the TMD/graphene system and the results here. The large anisotropy of spin lifetime in TMD has a similar physical origin based on the spin-valley locking effect as in the graphene/TMD system. We have cited the papers suggested by the reviewer in an expanded discussion of anisotropy and spin-valley locking in the revised manuscript on page 9, in addition to highlighting the anisotropy in the revised abstract and introduction.

“Again the temperature dependence found in the simulations of Xu et al could be seen as contradicting the analogy, but the same argument as discussed for graphene is likely to apply, mainly that the change in temperature is concomitant with a varying strength of SOC fields active in the ab-initio calculations.”

As we discussed earlier, the spin-phonon relaxation time decreases with increasing temperature, even for the DP mechanism case. The characteristic inverse relation with momentum relaxation time, or direct relation with relaxation rate, works for the *extrinsic* scattering rates due to impurities that are roughly temperature independent. We showed this for MoS₂ and GaN, but not for free-standing graphene, clearly distinguishing DP from non-DP cases. We have clarified this point more clearly in the revised manuscript on page 8, including in the revised abstract, to highlight this critical insight.

“I hope the authors will admit that if they revise the manuscript adding the ingredients above, their contribution will have an even larger impact in the field of spin transport in nanomaterials.”

Once again, we greatly appreciate the reviewer’s suggestions to gainfully expand the scope and utility of our work, and we have incorporated all of them in the attached manuscript as detailed above.

REVIEWERS' COMMENTS:

Reviewer #1 (Remarks to the Author):

The revised version is ready for publication